# A Novel Terahertz Detector Based on Asymmetrical FET Array in 55-nm Standard CMOS Process

**DOI:** 10.3390/ma15196578

**Published:** 2022-09-22

**Authors:** Yaxuan Liu, Xin Zhang, Jingye Sun, Ling Tong, Lingbing Kong, Tao Deng

**Affiliations:** 1School of Microelectronics, Tianjin University, Tianjin 300072, China; 2School of Electronic and Information Engineering, Beijing Jiaotong University, Beijing 100044, China; 3Beijing Microelectronics Technology Institute, Beijing 100076, China

**Keywords:** terahertz detector, CMOS, asymmetric MOSFET, grating structure

## Abstract

This paper reports a novel, one-dimensional dense array of asymmetrical metal-oxide-semiconductor field-effect-transistor (MOSFET) THz detector, which has been fabricated in GlobalFoundries 55-nm CMOS technology. Compared with other technologies, the Si-based complementary metal-oxide-semiconductor (CMOS) dominates in industrial applications, owing to its easier integration and lower cost. However, as the frequency increases, the return loss between the antenna and detector will increase. The proposed THz detector has a short-period grating structure formed by MOSFET fingers in the array, which can serve as an effective antenna to couple incident THz radiation into the FET channels. It not only solved the problem of return loss effectively, but also greatly reduced the detector area. Meanwhile, since the THz signal is rectified at both the source and drain electrodes to generate two current signals with equal amplitude but opposite directions, the source drain voltage is not provided to reduce the power consumption. This leads to a poor performance of the THz detector. Therefore, by using an asymmetric structure for the gate fingers position to replace the source drain voltage, the performance of the detector in the case of zero power consumption can be effectively improved. Compared with the symmetrical MOSFET THz detector, *R_v_* is increased by 183.3% and *NEP* is decreased by 67.7%.

## 1. Introduction

THz radiation (0.1–10 THz), which lies in the electromagnetic spectrum between microwave and infrared, shows various potential applications for imaging, security, spectroscopy technology and many others owing to its unique properties [1]. Since the theoretical consideration of plasma waves in a two-dimensional (2D) electron channel was proposed in [2], THz detectors based on the plasma theory have been extensively developed [3,4,5]. In order to achieve high-performance THz detectors, some applications have already been implemented using dedicated semiconductor technologies such as SiGe, GaAs, InP and InGaAs [6,7,8,9] or new materials such as photoelectric crystals [10] and two-dimensional (2D) materials [11]. Compared with the above technologies, the Si-based complementary metal-oxide-semiconductor (CMOS) technology dominates in industrial applications [12,13,14,15] due to its excellent advantages of low cost, low power consumption and large-scale integration of circuits by using readout electronics and on-chip signal processors. 

Among the published THz detectors, bolometers [16], Schottky barrier diodes (SBD) [8] and antenna-coupled metal-oxide-semiconductor field-effect-transistor (MOSFET) detectors [17], are used commonly. However, bolometer detectors exhibit a slow response speed. SBD detectors are difficult to integrate. Antenna-coupled MOSFET detectors show poor performance at high frequencies, since the return loss between the antenna and the detector will increase as frequency increases, due to the existence of antenna. In order to solve the above problems, the impedance matching structures have been used to reduce the return loss [18,19], hence the new antenna structures [20,21] or asymmetric MOSFET structures [22,23] have been used to improve the performance of the THz detector. Although these approaches can solve some problems, the impedance matching is very difficult to realize in high-frequency, and the new structure antennas are likely to further increase the detector area.

To solve the above problems, a room temperature 2.58 THz detector is proposed and demonstrated in GlobalFoundries 55-nm CMOS technology for the first time in this paper. It is composed of a short-channel asymmetrical MOSFET array, and the mechanism is plasma detection. The fingers in a separate short-channel FET array form a grating structure, serving as an effective antenna to couple the THz radiation into the FET channels [24]. This not only solves the high-frequency impedance matching problem, but also greatly reduces the area of the THz detector, since the fingers function as the antenna. In addition, the asymmetrical position of the gate fingers in the FETs can also break the shielding by changing the electrical balance in the channel [25]. The experimental results have shown that the performance of asymmetric detector is much better than symmetric detector. Hence, the short-channel asymmetric MOSFET array THz detector demonstrated a high *R_v_*, low noise equivalent power (*NEP*) and high response speed.

## 2. Materials and Methods

The diagram of the MOSFET array THz detector is shown in Figure 1a. Individual MOSFETs are densely arranged in the array to form a grating structure by using the fingers. The grating structure that acts as an effective matched antenna can couple the external THz radiation into MOSFET channels to excite plasmons, which can induce the response current in the channel. Therefore, the detector operating at a non-resonant mode can detect the THz radiation power by measuring the voltage at the drain or source terminal. Although MOSFET is not a two-dimensional device, the electric field provided by the gate accumulates a very thin layer of carriers (channel) below the gate oxygen layer because of the 0 V source-drain voltage. Due to the limitation of the electric field in the vertical direction (gate voltage), the movement of carriers in the channel can only move on the surface of the active region. Therefore, when analyzing the plasma in the MOSFET channel, it can be simplified into a two-dimensional model. Based on the two-dimensional electron channel, the plasma formula can be described with the two-dimensional electron flow [26].
(1)$$\frac{\partial v(x,t)}{\partial t}+v(x,t)\frac{\partial v(x,t)}{\partial x}+\frac{v(x,t)}{\tau }+\frac{e}{m^{*}}E(x,t)=0$$
(2)$$e\frac{\partial }{\partial t}n(x,t)-\frac{\partial }{\partial x}j(x,t)=0$$
where *E*(*x*,*t*) is the in-plane electric field depending on the time *t* and coordinate *x* in the 2D electron system, *τ* is the electron momentum relaxation time, *e* and *m** are the electron charge and the electron effective mass, respectively, *j*(*x*,*t*) = −*ev*(*x*,*t*)*n*(*x*,*t*) is the density of the induced electric current, *n*(*x*,*t*) and *j*(*x*,*t*) are the hydrodynamic electron density and velocity in the 2D electron channel.

As shown in Figure 1b, each MOSFET is coupled electromagnetically in the array, but they are uncoupled electronically since the MOSFETs are isolated with each other. Therefore, the response current *I* generated by each MOSFET is independent in the array. The THz radiation is uniform and the radiation area is large enough to ensure that each MOSFET in the array can receive the same amount of THz radiation. Meanwhile, each MOSFET in the array has the same gate bias and the common source and drain. The response currents generated by every MOSFET added together to produce the total output current *I*_total_. In principle, the response current *I* generated by each MOSFET is identical. 

The generated *I* in each MOSFET channel is given by the Fourier transform [27]:(3)$$\delta j_{0}=2\sigma_{0}\frac{e\gamma }{m^{*}}\sum_{q}\frac{q{\left|E_{q}\right|}^{2}}{\left(\omega_{0}-qv_{0})[{\left(\omega_{0}-qv_{0}\right)}^{2}+\gamma^{2}\right]}$$
where *σ*_0_ = *e*^2^*N*_0_*τ*/*m*^*^ is the conducting property of the plasmons in the channel, *γ* = 1/*τ*, *E_q_* is the Fourier harmonic of the total self-consistent in-plane electric field of frequency *ω*_0_ with wave vector *q* = 2*πn*/*l*, *n* = 0, ±1, ±2, …, and *l* is the grating period.

Except for the shielding by gate bias, THz radiation rectified at the drain or source electrodes are crucially dependent on how the THz signal is fed to the channel. The two possibilities feature an opposite sign of the rectified current. This competition will reduce the performance of the detector [28]. So, the THz detector demands a design for which only one of the two mechanisms is dominant. This can be achieved by an asymmetrical structure or biasing conditions. 

According to Equation (3), a symmetrical unit cell without DC electron drift in the 2D electron channel causes |*E_q_*| and |*E_−q_*| to be equal. The Fourier-harmonics of the electric field, which have the wave vectors *q*th of opposite signs, have equal amplitudes and hence the total output current is zero. Therefore, the strong asymmetrical position of the gate fingers can lead to different |*E_q_*| and |*E*_−*q*_| to improve *R_v_* of the THz detector. *R_v_* also can be further improved by changing the drain-source bias (*V*_ds_) because the difference between |*E_q_*| and |*E*_−*q*_| also depends on the local electron velocity *v*_0_, which is equal to −*eτV*_ds_/*m*^*^. Meanwhile, the gate bias (*V*_gs_) can increase the local electron density *N*_0_ to improve the *R_v_*.

In addition to *R_v_*, noise is another key performance indicator for the THz detector. The MOSFET channel material is resistive, so the major noise comes from the thermal noise. The theoretical estimation of the room temperature Johnson-Nyquist voltage noise floor is expressed as (4*k*_B_*TR*_total_)^1/2^, where *k*_B_ is the Boltzmann constant, *T* is the temperature, and *R*_total_ is the measured resistance [29]. At the same temperature, the noise is only dependent on the resistance between the source and drain. The resistance of the channel is related to the gate width (*W*) and length (*L*). The higher value of *W/L* is the lower value which the channel resistance presents. As the same *W*, the short channel MOSFET shows better noise characteristics than long channel MOSFET. Compared with the HEMT technology, the CMOS process has smaller *L*. The detector employing parallel MOSFETs to realize the grating structure can achieve smaller *L* compared with grating-gate detector. Thus, the short-channel asymmetrical MOSFET array THz detector also exhibits lower noise. 

In order to further verify the absorption performance of the grating structure, Figure 2 shows that the absorption rates of the grating structure are controlled by changing the length and width of the fingers. The absorption rates were simulated by the high frequency structure simulator (HFSS). It can be seen from Figure 2 that changing the length and width of the grating structure will affect its absorption capacity. The absorptivities of grating structures with five different sizes was simulated (size parameters are shown in the Figure 2). After comparing the simulation results, the optimized values for the length and width fingers were *L* = 2 µm and *W* = 25 µm, respectively. It can be obtained from the simulation results that the grating structure has a stronger absorption capacity at 2.58 THz. 

The schematic diagram for the proposed detector is shown in Figure 1a, which was fabricated on top of Si wafer just using n-/p-well. The active regions of MOSFETs are formed on silicon by doping. The polysilicon gates with a thickness of 94 nm and active regions are separated by the gate oxygen layer. The active regions on both sides of the gates are connected with the upper metal (thickness of 220 nm) through vias with a thickness of 216 nm, forming the source and drain electrodes. Finally, multiple MOSFETs are connected in parallel to the top pad.

Figure 3a,b shows the die micrograph of the symmetrical and asymmetrical MOSFET arrays that are fabricated in 55-nm CMOS technology, where the finger length *L*_g_ of each MOSFET is 2 µm. This small size can effectively reduce the noise of the detector. Meanwhile, in order to improve the response current, six MOSFETs were used in the array. For the symmetrical detector, the slits *L* for the gate-source and gate-drain are 500 nm. The size of this detector is used for performance comparison with the asymmetrical detector. For the asymmetrical detector, the gate finger is placed in a position *L*_1_ = 300 nm from the source finger and *L*_2_ = 700 nm from the drain finger, which provides the necessary asymmetry to further improve *R_v_*.

## 3. Results and Discussion

The measurement setup is shown in Figure 4. The THz radiation is generated by a quantum cascade laser (QCL) operating at 2.58 THz with a maximum power of 1.8 mW. The focused beam with a diameter of ~100 µm was collimated and focused by two parabolic optical mirrors. The THz source and the lock-in amplifier were simultaneously modulated by a signal generator for synchronization. The THz detectors were mounted on an x–y–z stage and positioned at the focus point of the THz beam. The received THz power was measured by a terahertz probe (Ophir 3A-P-THz, Jerusalem, Israel). The output voltage was measured through a lock-in amplifier, and the voltage is supplied by a power supply.

The electrical transfer curves of the symmetrical and asymmetrical detectors are shown in Figure 5. According to the results, it can be seen that the asymmetrical structure will not degrade the basic switch characteristics of standard MOSFET. This is mainly because both the gate electrode length/width and the distance between the source and drain electrodes have not been changed, the output characteristics of MOSFET show insignificant change and can be ignored. When the gate voltage (*V*_gs_) of the asymmetrical MOSFET is greater than the threshold voltage, there is a maximum value of slope (transconductance, *g*_m_) at about *V*_gs_ = 0.45 V. Since the performance of the detectors is mainly affected by *V*_gs_, a larger source drain voltage (*V*_ds_) cannot significantly improve the performance. In addition, excessive *V*_ds_ will increase the power consumption of the detector. Therefore, a smaller *V*_ds_ was adopted in the output characteristics (<100 mV), so that the MOSFET operated in a linear region. As shown in Figure 5, a linear relationship between the source drain current (*I*_ds_) and the *V*_ds_ is obtained, indicating that the MOSFETs have good performance.

The output voltage (Δ*U*) for the asymmetry detector under the illumination of THz radiation with various actual power (*P_actual_*) (13.38, 53.50, 101.66, 148.47, 189.94 and 224.71 µW) is shown in Figure 6a. The performance of the symmetrical detector is not shown in Figure 6a, due to its response being too low, and there is no response when the actual power is lower than 224.71 W. The Δ*U* for the symmetry detector is 1.53 µV with the *P_actual_ =* 224.71 W. Much less than the Δ*U* for the asymmetric detector (4.18 µV). With the increase in incident THz radiation power, Δ*U* of the THz detector changes linearly, which meets the requirements of a THz power detector. The dotted fitting line in the figure also implies excellent linearity between Δ*U* and the incident power.

Then, the voltage *R_v_* was measured and calculated by the following formula [20]:(4)$$R_{v}=\frac{\Delta U\times S_{radiation}}{P_{total}\times S_{detector}}$$
where Δ*U* is the amplitude of the response voltage, *P_total_* is the total power of the THz source, *S_radiation_* is the radiation beam spot area and *S_detector_* is the area of the detector.

Here, the actual area of the detector is 1050 µm^2^, the area of the radiation beam spot is 7850 µm^2^, and the actual maximum received THz power on the detector is calculated as *P_receive_* = 224.71 µW. Figure 6b illustrates the *V*_gs_-dependence *R_v_* at 2.58 THz for symmetrical and asymmetrical detectors under *V*_ds_ = 0 V and *P_receive_* = 224.71 µW. With the increase in *V*_gs_, *R_v_* of asymmetry and symmetry detectors increase firstly and then decrease, where the maximum values are obtained at *V*_gs_ = 0.25 V and *V*_gs_ = 0.45 V, respectively. This is consistent with the highest *R_v_* when the transconductance is at a maximum. As the maximum transconductance, it indicates that the same gate voltage change will lead to the largest source drain current. As shown in Figure 1b, the grating structure acts to an antenna to couple THz radiation to the MOSFET gates, which is equivalent to adding *U*_a_ to the gates. Therefore, when the transconductance is maximum, the *R_v_* is maximum. It is also found that *R_v_* of the asymmetrical detector is obviously higher than that of the symmetrical detector (2.83 times). These characteristics are in good agreement with the detection theory described above.

Figure 7 shows the relationship between the output voltage of the asymmetry detector and the modulation frequency of the THz source. The responsivities at different modulation frequencies implies the bandwidth and response speed of the detector. The results show that when the modulation frequency is within 1 kHz, the output voltage of the asymmetry detector is stable. When the modulation frequency is greater than 1 kHz, the output voltage drops significantly. Meanwhile, the 3 dB bandwidth (output voltage drops by 50%) is about 3 kHz. This indicates that the detector shows stable output and good performance within 3 kHz bandwidth. As the modulation frequency increases to 10 kHz, the asymmetry detector still has a good signal-to-noise ratio. This shows that the detector is able to achieve a relative high response speed of 0.1 ms.

Further, the *NEP* is evaluated from the experimental data of *R_v_* and the noise spectral density *S_v_*, which is expressed as
(5)$$NEP=\frac{S_{v}}{R_{v}}$$
where *S_v_* is the detected noise voltage of the detector measured by a dynamic signals analyzer (Stanford Research Systems SR780, Stanford, CA, USA). The noise spectral density of the asymmetry detector is shown in Figure 8a. The corner frequency of 1/*f* noise is about 2 kHz. Figure 8b shows the asymmetry detector *V*_gs_-dependence of *NEP* at a modulation frequency of 1 kHz. When *V*_gs_ = 0.45 V, *NEP* has a minimum value of 2.55 nW/Hz^0.5^. However, due to the influence of 1/*f* noise, the noise is large at 1 kHz. While, at 10 kHz modulation frequency, although *R_v_* is low, the minimum *NEP* is about 1.29 nW/Hz^0.5^, since the 1/*f* noise is avoided.

## 4. Conclusions

In this paper, a novel asymmetrical MOSFET array plasma detector have been demonstrated and shown to have a high *R_v_* and an excellent *NEP* at 2.58 THz. The grating structure formed by fingers in MOSFET array can be considered as an effective THz antenna. This structure avoids the use of an antenna, thus solving the impedance matching problem between the antenna and the detector at high frequencies. Moreover, this structure also reduces the area of the detector. With the increase in frequency, the performance of plasma THz detectors gradually deteriorate. The asymmetrical position of the gate fingers can break the shielding by changing the electrical balance in the channel. This enables the improvement in *R_v_* of the THz detector on the premise of zero power consumption. The experimental results show that at 2.58 THz, the maximum *R_v_* and minimum *NEP* for the symmetrical MOSFET THz array detector are 6.78 mV/W and 7.88 nW/Hz^0.5^, respectively, while, for the asymmetrical detector, these two values are 18.9 mV/W and 2.55 nW/Hz^0.5^, where *R_v_* is increased by 183.3% and *NEP* is decreased by 67.7%. When compared with the existing HEMT grating-gate detectors, the proposed MOSFET array detector employed a shorter channel and parallel structure. The short channel MOSFET could reduce the channel resistance, thereby reducing the thermal noise, and effectively optimizing the *NEP* of the detector. The use of parallel structure would increase the output voltage of the detector, which is believed to increase the application range of the detector. The detector was fabricated using standard Si-based CMOS technology, which makes it easier to integrate with other processing circuitry, including transimpedance amplifiers, voltage limiters, clock and data recovery circuits. In sum, the detector proposed in this paper exhibit good performance and low cost in design and fabrication, showing broad application prospects.

## Figures and Tables

**Figure 1 materials-15-06578-f001:**
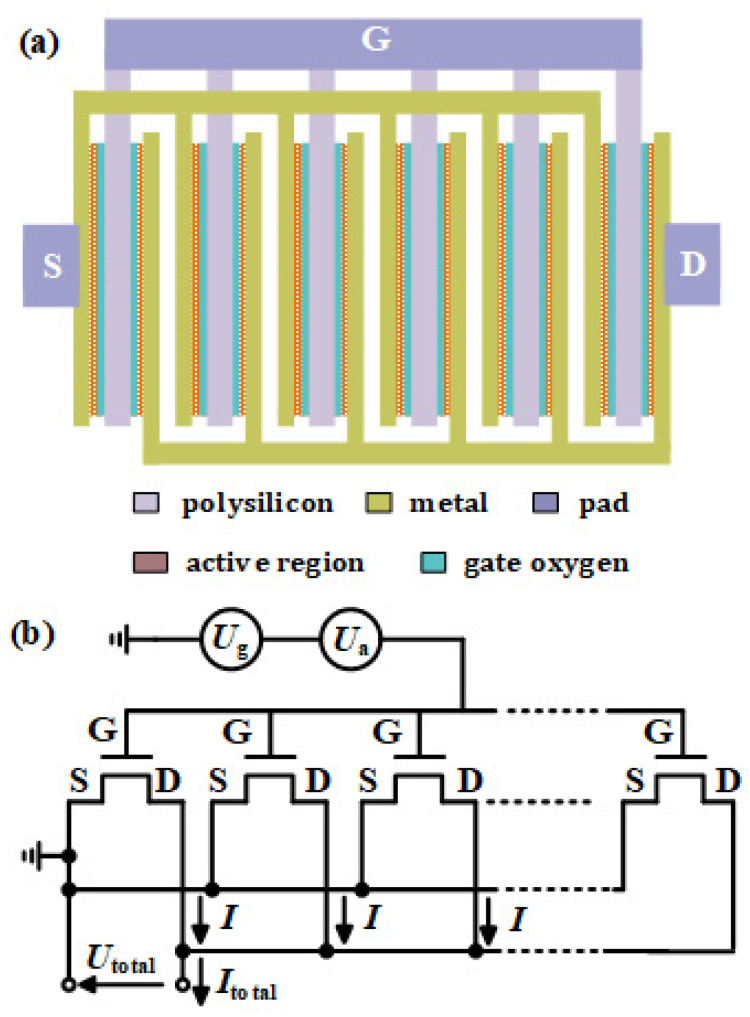
MOSFET array terahertz detector: (**a**) schematic diagram and (**b**) equivalent circuit model.

**Figure 2 materials-15-06578-f002:**
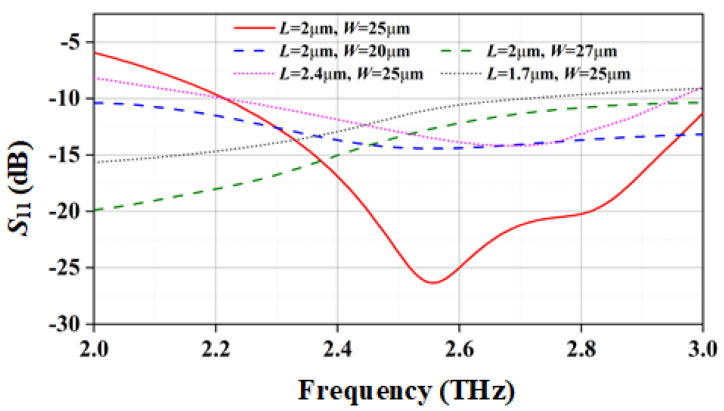
Simulated return loss of grating structure.

**Figure 3 materials-15-06578-f003:**
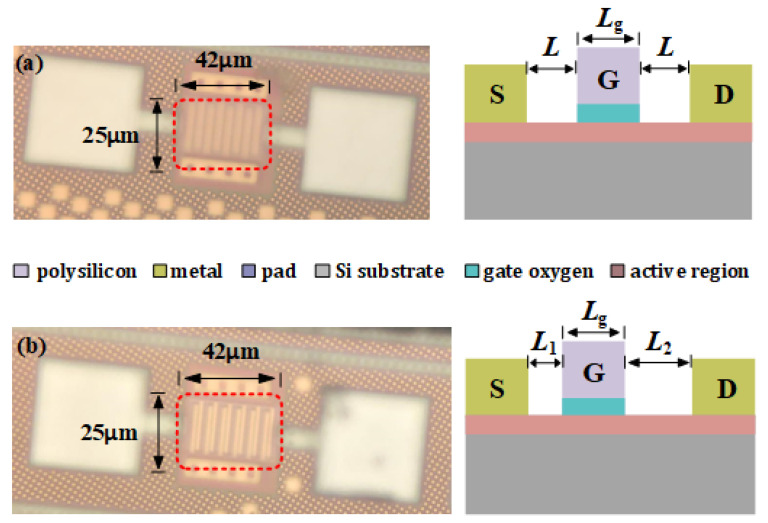
(**a**) Die micrograph and cross-section of the symmetrical detector; (**b**) die micrograph and cross-section of the asymmetrical detector.

**Figure 4 materials-15-06578-f004:**
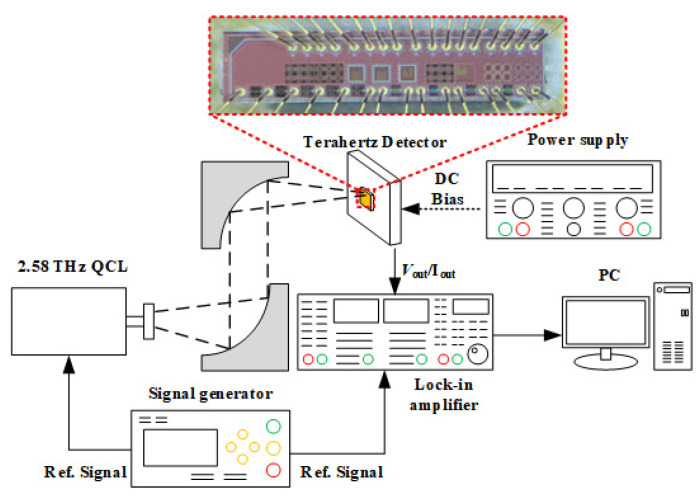
The THz measurement setup.

**Figure 5 materials-15-06578-f005:**
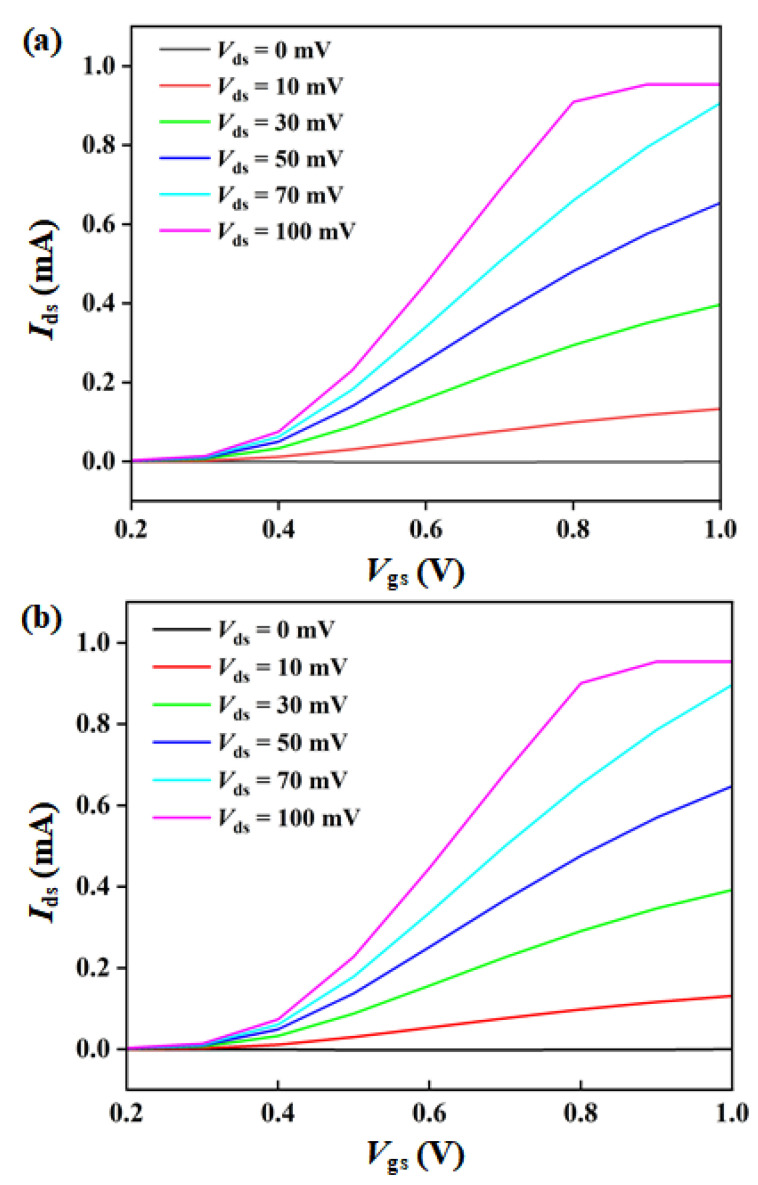
Output characteristic curves as a functions of the *V*_gs_ for various *V*_ds_: (**a**) symmetrical detector and (**b**) asymmetrical detector.

**Figure 6 materials-15-06578-f006:**
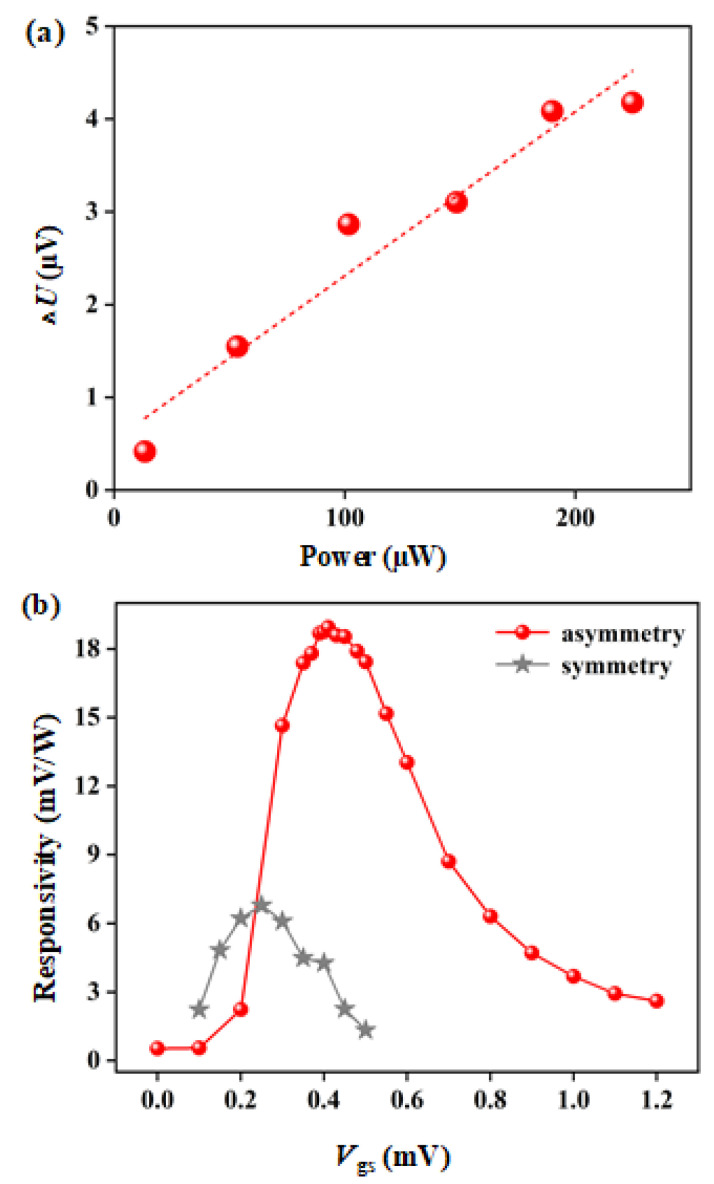
(**a**) The output voltage with different incident terahertz power under *V*_gs_ = 0.43 V and *V*_ds_ = 0 V conditions; (**b**) the asymmetry/symmetry detectors *V*_gs_-dependence responsivity at *V*_ds_ = 0 V.

**Figure 7 materials-15-06578-f007:**
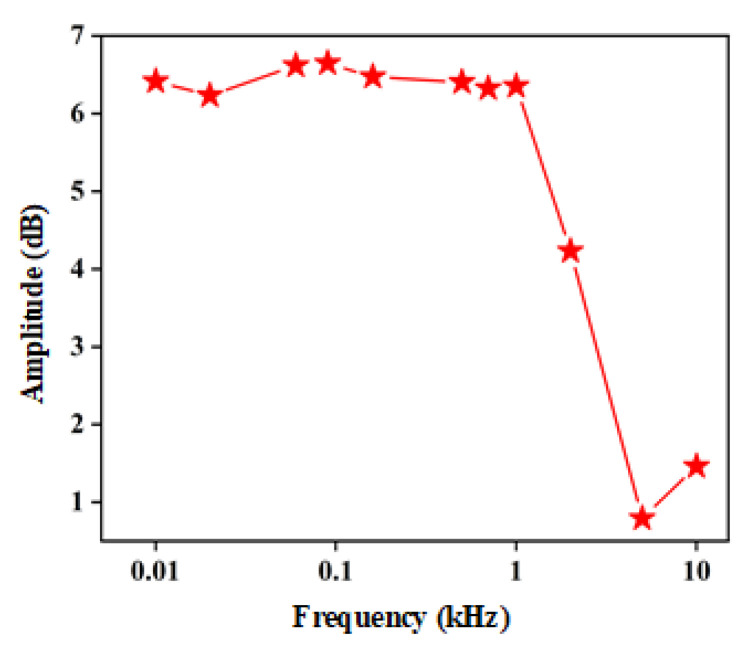
Output voltage of the asymmetry detector versus modulation frequency.

**Figure 8 materials-15-06578-f008:**
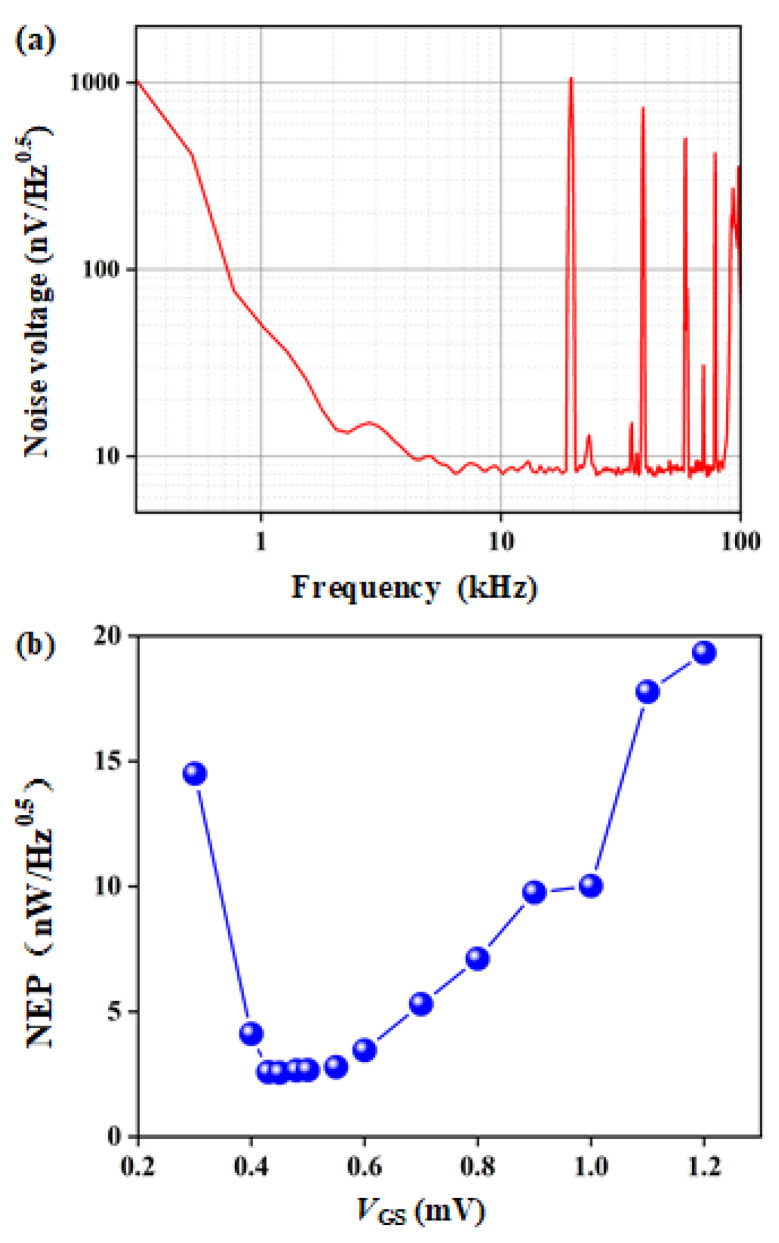
(**a**) Noise spectral density of the asymmetry detector at *V*_gs_ = 0.43 V and *V*_ds_ = 0 V; (**b**) the asymmetry detector *V*_gs_-dependence *NEP* at a modulation frequency of 1 kHz.

## Data Availability

Not applicable.

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
