# Peer review of "A Novel Terahertz Detector Based on Asymmetrical FET Array in 55-nm Standard CMOS Process"

_materials, 2022, doi:10.3390/ma15196578_

Round 1

Reviewer 1 Report

The authors shows a terahertz detector based on asymmetrical FET. The authors shows an good FET output performance based on the asymmetrical structure. There are some issues that can be clarified as

1. The plasma formula eq.(1) and (2) is valid in 2D device. As the FET used in this work manuscript is not 2D device, please explain and show the adequacy for using the 2D theory in this work.

2. For the CMOS description(line 66 and hereinafter), there should be an integrated n-channel and p-chnnel FET. As the device shown in Fig.1 is the MOS-FET only, please shows the consistent discussion between CMOS process and MOS-FET. 

3. Please explain the frequency difference for 2.58 THz (Line 144) and 2.52 THz(Fig.4)  

4. For the setup shown in Fig 4, is there any hardware line between the laser and lock-in amplifier for synchronization ? 

 5. As the space L1 and L2 in Fig. 3 take effect of basic FET performance(Ids, Vds Vgs), please shows the basic characters of the symmetrical and non-symmetrical MOS-FET. 

Reviewer 2 Report

First of all, congratulations to the authors for the research work presented in this journal. This paper describes a new terahertz detector based on an asymmetric FET array in a standard 55nm CMOS process. 

The abstract does not clearly indicate what the authors' contributions have been to the study problem addressed in the document. In this way, the advances, results, conclusions, and importance of the research carried out are not appreciated. It is recommended to rewrite the summary indicating the main contributions of the authors, as well as the conclusions obtained. As for the results, it is sufficient to indicate a brief summary of them in the abstract. All these mentioned details appear throughout the document.

The introduction contains the state of the art associated with the study problem. In this case, the main contributions of the authors to the problem analyzed have not been very clear, nor have the advantages and disadvantages compared to other research carried out. As a suggestion, perhaps the development of a table (showing the contributions of the most important references) can give greater clarity to the state-of-the-art. It is also recommended to increase the number of bibliographic references in the document. This will help to delve into the state-of-the-art.

On the other hand, the document is technically sound, since it contains a small analysis of the state-of-the-art related to the problem addressed. It also incorporates different mathematical equations and diagrams associated with the developed model. It also includes some estimated results that support the analysis shown. These results are compared with the experimental results obtained.

The concepts are presented comprehensively. The different figures, tables, diagrams, and schemes facilitate the understanding of the contents presented by the authors in the document. Likewise, the results obtained support the comments made by the authors. All this makes it easier for the reader to follow the document.

The simulation results are supported by the experimental results.

The conclusions are too short. Also, most of the text incorporated in this section is more discussion-oriented. It is recommended to rewrite this section and includes if the author considers it appropriate, a section dedicated to the discussion of the results described. On the other hand, as mentioned above, the number of bibliographical references provided by the author is insufficient. It is advisable to increase their number to improve the state-of-the-art.

Reviewer 3 Report

The authors described the manuscript entitled"A Novel Terahertz Detector Based on Asymmetrical FET Array 2 in 55-nm Standard CMOS Process"  and discussed a novel one-dimensional dense array of asymmetrical CMOS field-effect-transistor (FET) THz detector which has been fabricated in Globalfoundries 55-nm CMOS technology in a well-organized way. Paper is useful to the scientific community however some points must be explained before acceptance:

1. Novelty and objectives of work should be clearly mentioned.

2. Why does the asymmetrical structure not degrade the basic switch characteristics of standard CMOS FET?

3. Why the source-drain current (Ids)   increases linearly with the increase of the source-drain voltage (Vds)? Explain.

4. When the modulation frequency is greater than 1 kHz, the output voltage drops significantly, why? Explain.

5. Include the latest references 2020,2021,2022.

Reviewer 4 Report

The authors present the fabrication and characterization of a terahertz detector based on asymmetrical FET array in 55-nm standard CMOS process. To show the advantages of an asymmetrical FET array against the symmetrical counterpart, they also fabricate the symmetrical detector. However, they do not compare the measurements of both approaches, except in Figure 6, where measurements of the responsivity of both types of detectors are compared for a single frequency and power. Therefore, I would recommend carrying out a deeper comparison among both detectors to bring to light the advantages and shortcomings of each of the structures.  

The manuscript is not clear written. Think that it would be benefited from a language edition. I strongly recommend a deep revision of it.

My main concern is that the authors relate the detector responsivity with the transistor transconductance, and they affirm that the maximum responsivity of the detector is achieved at the maximum transconductance. However, when a device is operated as a detector, its non-linear characteristics is exploited.  The transconductance is the derivative of the current with the gate voltage and is a figure of merit used for linear applications such as amplifiers. Can the authors explain why a maximum transconductance should mean a maximum responsivity?

The authors must discuss the differences of their asymmetric detector against the approaches reported in the references, otherwise the novelty will not be clear.

A fabrication section must be included. There is no information about the prototype fabrication.

Measurements of the symmetrical device must be included in Figs. 5, 6a, 7 and 8 for a fair comparison. Please, indicate the power considered in Fig. 6b. For a different power, will the responsivity curves be the same?

All the curves shown in Fig. 2 must be discussed.

The fabrication section is too short. Please, give all the details about the prototype fabrication.

How do the authors measure the distance between gate-source and gate-drain for the symmetrical (500nm) and asymmetrical (300-700nm)?

The theoretical explanation (basically the whole Section 2) and Eqs. (1)-(3) are useless, I suggest removing it. For example, the following paragraph: <<At the same temperature, the noise is only related to the resistance between the source and drain. The resistance of channel is related to the gate width (W) and gate length (L). The higher value of W/L, the lower channel resistance. As the same W, the short channel FET has better noise characteristics than long channel FET. Compared with the HEMT technology, CMOS process has smaller L. The detector using the parallel CMOS FETs to realize the grating structure can achieve smaller L compared with grating-gate detector. Thus, the short-channel asymmetrical CMOS FET array THz detector also has better noise.>> is not acceptable for a scientific manuscript. It is a very simple analysis that does not take into account possible physical effects such as: short-channel effects, self-heating effects that increase the temperature and thus the noise, velocity saturation effects, etc.  Indeed, there are many noise sources that are not considered in the draft nor commented.

Some sentences that must be explained:

§  As the frequency increases, the return loss between the antenna and detector will increase.” How do the authors define return losses? Can the authors give a reference that support such statement?

§  the long grating-gate FET channel will cause large voltage drop.” Why? Where is the voltage drop produced (gate-source, drain-source)? How much is it a “large voltage drop”?

§  In addition, the asymmetrical position of the gate finger in the FET can also break the shielding by changing electrical balance in the channel”. Can the authors demonstrate that the asymmetrical position of the gate fingers breaks the shielding? I cannot deduce such effect from the data reported in the draft. If so, what kind of measurements have the authors carried out to show that?

§  the asymmetrical structure will not degrade the basic switch characteristics of standard CMOS FET.” To demonstrate that, the authors must show the transfer characteristics in logarithmic scale.

§  The long channel grating-gate CMOS FET is difficult to be applied in high frequency due to its low electron mobility”. Therefore, the short-channel grating-gate CMOS FET is also difficult to be applied in high-frequency, given that the electron mobility is an intrinsic property of the material, isn’t it?

Round 2

Reviewer 1 Report

The authors have done the necessary correction. This manuscript can be accepted.

Reviewer 2 Report

First of all, congratulations to the authors for the research work carried out.

The authors have incorporated the different observations and comments made by the reviewers. In this way, they have rewritten the introduction and the conclusions. They have also improved the state-of-the-art and incorporated a greater number of bibliographical references in the document.

All these details have made it possible to improve the paper.